# Antibiotic Knockdown of Gut Bacteria Sex-Dependently Enhances Intravenous Fentanyl Self-Administration in Adult Sprague Dawley Rats

**DOI:** 10.3390/ijms24010409

**Published:** 2022-12-27

**Authors:** Michelle Ren, Shahrdad Lotfipour

**Affiliations:** 1Department of Pharmaceutical Sciences, School of Pharmacy and Pharmaceutical Sciences, University of California Irvine, Irvine, CA 92697, USA; 2Department of Emergency Medicine, School of Medicine, University of California Irvine, Irvine, CA 92697, USA; 3Department of Pathology and Laboratory Medicine, School of Medicine, University of California Irvine, Irvine, CA 92697, USA

**Keywords:** addiction, microbiota, opioids, reward, reinforcement, dysbiosis

## Abstract

Communication between the brain and gut bacteria impacts drug- and addiction-related behaviors. To investigate the role of gut microbiota on fentanyl reinforcement and reward, we depleted gut bacteria in adult Sprague Dawley male and female rats using an oral, nonabsorbable antibiotic cocktail and allowed rats to intravenously self-administer fentanyl on an escalating schedule of reinforcement. We found that antibiotic treatment enhanced fentanyl self-administration in males, but not females, at the lowest schedule of reinforcement (i.e., fixed ratio 1). Both males and females treated with antibiotics self-administered greater amounts of fentanyl at higher schedules of reinforcement. We then replete microbial metabolites via short-chain fatty acid administration to evaluate a potential mechanism in gut-brain communication and found that restoring metabolites decreases fentanyl self-administration back to controls at higher fixed ratio schedules of reinforcement. Our findings highlight an important relationship between the knockdown and rescue of gut bacterial metabolites and fentanyl self-administration in adult rats, which provides support for a significant relationship between the gut microbiome and opioid use. Further work in this field may lead to effective, targeted treatment interventions in opioid-related disorders.

## 1. Introduction

The number of opioid-related deaths is continuing to rise with the most recent spike in deaths driven by fentanyl [1,2]. This highlights the urgency to identify underlying mechanisms contributing to fentanyl abuse and addiction. There is a clear association between opioids and gut health given the gastrointestinal (GI) side effects from opioid use (e.g., constipation, nausea) due to the widespread distribution of opioid receptors throughout the GI tract [3,4]. The trillions of microbes that reside in the intestines are known as gut bacteria (also called microbiota or flora) and collectively make up an organism’s gut microbiome [5]. In this study, we show communication between gut bacteria and the brain as a potential mechanism underlying fentanyl intravenous self-administration (IVSA) in adult Sprague Dawley rats.

Bidirectional communication through the gut-brain axis has been demonstrated to mediate neuropsychiatric disease [6,7]. The gut and brain are physically connected via the vagus nerve [8], which has been shown to mediate behavior in mouse models of anxiety and depression [9,10,11]. Gut-brain communication also occurs biochemically through hormones, neurotransmitters, immune signaling, and microbial metabolites [12,13,14]. Evidence that the brain and gut microbiota interact with each other has been established in a variety of studies. Animals raised and maintained with no microbes (i.e., germ-free) compared to conventional animals demonstrate differences in anxiety- and depression-like behaviors, locomotor activity, gene expression, microglia, and neurogenesis [15,16,17,18,19]. Additionally, the administration of specific bacterial strains, fecal microbiota transplantation, or antibiotics distinctly impacts behavior, the brain and spinal cord, and gut health [9,11,20,21,22,23].

Gut bacteria influence neural circuits and behaviors that are markedly associated with addiction, including reward, tolerance, and withdrawal [24,25,26,27,28,29,30]. While alterations in gut microbiota directly affect opioid-related behaviors, opioid exposure also alters the diversity and/or composition of the host gut microbiome [26,31,32,33,34,35,36,37]. We previously demonstrated the impact of fentanyl self-administration on gut microbiota [33]. To highlight the bidirectional communication between the brain and gut microbiota, our present study aims to evaluate the role of gut microbiota on fentanyl self-administration. As depletion of gut bacteria via oral antibiotic treatment has been shown to enhance sensitivity to cocaine reward and disrupt opioid reward [24,28], we test the hypothesis that knocking down gut bacteria will enhance fentanyl self-administration due to dysregulated reward processing. We assess alpha diversity from fecal samples of water- and antibiotic-treated animals to confirm that the selected antibiotic doses and duration of treatment significantly deplete gut bacteria. Subsequently, we administer short-chain fatty acids (SCFAs) to bacteria-depleted rats to examine the impact of gut microbial repletion on fentanyl self-administration. SCFAs are the main metabolites produced by bacterial fermentation of dietary fiber in the GI tract, and prior work demonstrated SCFA treatment in antibiotic-treated animals restores behavioral responses akin to controls [24].

The lack of a normal, healthy gut microbiome is understood to contribute to depression and anxiety [11,16,20,21,38,39,40], and accumulating evidence shows that imbalances in gut bacteria (i.e., gut dysbiosis) are also linked to opioid use disorder [41,42]. The antibiotic treatment in this study is a means to significantly deplete gut bacteria [24] and does not mirror clinical doses. The cited animal studies addressing the connection between drugs of abuse and gut microbiota evaluate drug reward using conditioned place preference. We use an intravenous model of self-administration on an escalating schedule of reinforcement to assess drug reinforcement and motivation. Evaluating the bidirectional relationship between gut microbiota and fentanyl reinforcement and reward contributes to the limited understanding of mechanisms mediating opioid dependence and abuse. Building upon this foundation may allow for the development of tractable treatment options for opioid use.

## 2. Results

### 2.1. One-Week Oral Antibiotic Treatment Significantly Depletes Gut Bacteria (and Depletion Is Maintained throughout Experiment)

We analyzed bacteria of fecal samples from a subset of males in our self-administration study. We evaluated the diversity of gut bacteria using the Shannon diversity index (richness and evenness) and the relative abundance of specific phyla and genera. We used a 2-way, repeated measures ANOVA to analyze differences in Shannon diversity between water- and antibiotic-treated rats before and after fentanyl self-administration to ensure that the selected antibiotic cocktail knocked down gut bacteria prior to the start of self-administration and that knockdown was maintained throughout the experiment. We found a significant reduction in Shannon diversity in antibiotic-treated vs. water-drinking controls both before (F(1,6) = 195.76, *p* < 0.0001) and after (F(1,6) = 116.18, *p* = 0.0001) self-administration (Figure 1). In addition to a main effect of treatment, we found a significant difference in Shannon diversity before and after fentanyl self-administration in water-treated animals (*p* = 0.01) (Figure 1).

### 2.2. Antibiotic Treatment Causes Phylum- and Genus-Level Changes in Gut Bacteria

To assess phylum-level changes in treatment groups before and after fentanyl self-administration, we ran a one-way, repeated measures ANOVA to identify differences in the relative abundance of *Firmicutes*, *Bacteroidetes*, *Proteobacteria*, *Tenericutes*, *Verrucomicrobia*, and *Actinobacteria*. We observed a significant increase in *Bacteroidetes* (*p* = 0.002) in antibiotic-treated animals after vs. before fentanyl self-administration but saw no timepoint differences in controls (Figure 2A). Further, we found a significant decrease in the percent relative abundance of *Firmicutes*, *Tenericutes*, and *Actinobacteria*, and a significant increase in *Bacteroidetes, Proteobacteria*, and *Verrucomicrobia* in antibiotic-treated animals compared to controls independent of fecal collection timepoint (before or after self-administration) (Figure 2B). This indicates that one week of antibiotic treatment is sufficient to change all six phyla evaluated here. Analysis at the genus level did not reveal the specific changes in Bacteroidetes in the antibiotic-treated group before vs. after fentanyl self-administration; however, we found a higher relative abundance of the genus *Prevotella* in control animals compared to those treated with antibiotics (Figure 3). We also found specific genus differences in the Firmicutes phylum when comparing controls and antibiotic groups (Figure 3). Animals drank the same amount of water and maintained the same weight regardless of the treatment group.

### 2.3. Antibiotic-Treated Males, but Not Females, Self-Administer More Fentanyl Than Controls at Fixed Ratio (FR) 1

We ran a 2-way ANOVA for each schedule of reinforcement to assess the role of treatment and sex on fentanyl self-administration at 1.25 μg/kg/infusion. As there was no difference in responding on any day at FR1, we analyzed a 2-day average. The last two days of FR1 (days 4–5 of self-administration) were selected, as animals may take time to acquire drug self-administration. At FR1, we found a significant interaction between treatment and sex (F(3,31) = 4.5, *p* = 0.04), thus data were analyzed separately for males and females. We observed a main effect of treatment on reinforced responding in males (*p* = 0.02) but not females (*p* = 0.78) (Figure 4A).

### 2.4. Antibiotic-Treated Animals Self-Administer More Fentanyl Than Controls at FR2, FR5, and Progressive Ratio (PR)

Subsequent 2-way ANOVAs showed a main effect of treatment at FR2 (F(3,31) = 3.1, *p* = 0.02), FR5 (F(3,31) = 3.1, *p* = 0.04), and PR (F(3,31) = 4.11, *p* = 0.008), with higher reinforced responses in antibiotic-treated animals, compared to water-drinking controls (Figure 4B,C). Data are collapsed by sex due to a lack of sex differences at these higher schedules of reinforcement. As the responses required to earn an infusion increases logarithmically on a PR schedule of reinforcement, the number of infusions self-administered is much lower on PR than FR. Thus, the scale displaying mean infusions is smaller in PR (Figure 4C) compared to FR1, FR2, and FR5 (Figure 4A,B).

### 2.5. Short-Chain Fatty Acid (SCFA) Supplementation Blunts Fentanyl Self-Administration in Antibiotic-Treated Animals at FR2 and FR5

Only males were used for the SCFA supplementation study, as the enhancement of fentanyl self-administration seen from antibiotic treatment was driven by males. A one-way ANOVA was run separately for all schedules of reinforcement to analyze differences in all 4 treatment groups on fentanyl self-administration at 1.25 μg/kg/infusion. We found a main effect of treatment at FR1 (F(3,34) = 4.52, *p* = 0.009), FR2 (F(3,34) = 6.57, *p* = 0.001), FR5 (F(3,34) = 3.58, *p* = 0.02), and PR (F(3,34) = 3.41, *p* = 0.02) (Figure 5). At FR1, animals treated only with antibiotics had higher reinforced responses compared to those treated with water only or water with SCFAs (Figure 5A). Additionally, treatment with antibiotics plus SCFAs resulted in higher reinforced responses at FR1 versus water-only controls (Figure 5A). At both FR2 and FR5, the same pattern of differences was observed: antibiotic treatment alone led to increased reinforced responses compared to all other treatment groups (water only, water with SCFA, and antibiotics with SCFA) (Figure 5A). These data show that the repletion of microbial metabolites via SCFAs restores self-administration similar to control animals at higher fixed ratio schedules of reinforcement. At PR, there was no significant difference in SCFA-supplemented animals treated with antibiotics compared to antibiotic treatment only, although this effect was trending (*p* = 0.08) (Figure 5B).

## 3. Discussion

Our findings highlight an important relationship between the knockdown of gut bacteria and fentanyl IVSA with an enhancement of self-administration driven by males. In addition, we find that at higher schedules of reinforcement (i.e., FR2 and FR5), SCFA supplementation in antibiotic-treated males decreases fentanyl self-administration compared to males treated with antibiotics, suggesting that microbial metabolites may mediate the reinforcing efficacy of fentanyl. A previous study found a similar trend in cocaine reward in mice with knockdown and later restoration of bacterial metabolites [24]. Our present study is consistent with prior work supporting a relationship between altered gut microbiota and opioid-related behaviors in rodents. While these prior groups have reported depletion of gut bacteria to decrease opioid tolerance and preference using non-contingent drug exposure [25,27,28,43], we show an increase in opioid self-administration. Although conditioned place preference and IVSA both measure drug-related behavior and learning, they are drastically different methods and yield conclusions with subtle differences in reward, motivation, and reinforcement. These discrepancies may also be potentially explained by differences in species (rats vs. mice), the opioid used (fentanyl vs. morphine), duration of drug exposure, route of administration (intravenous vs. subcutaneous or intraperitoneal), and method of administration (self-administration vs. experimenter-administered). Additional experimentation, particularly self-administration studies, will be necessary to interpret such inconsistencies.

Our findings are also consistent with clinical studies that establish an association between opioid use and subsequent gut dysbiosis [34,36,44,45,46]. We observed an increase in Bacteroidetes in antibiotic-treated males before and after fentanyl IVSA. The Firmicutes/Bacteroidetes (F/B) ratio plays a role in maintaining normal intestinal homeostasis, with a higher or lower ratio observed in obesity or inflammatory bowel disease, respectively [47]. However, the findings on the beneficial vs. harmful features of bacterial species within Bacteroidetes are mixed, as some studies suggest Bacteroidetes decrease inflammation and improve body composition [48], while others highlight associations between Bacteroidetes and metabolic diseases [49]. This may be due to the various genera that make up this phylum, although we did not find specific differences at the genus level of Bacteroidetes. We observed many genera that were decreased from antibiotic treatment compared to water treatment, which is expected considering the significant depletion of bacteria from our antibiotic cocktail.

One recent national database study found that opioids prescribed in combination with antibiotics in the hospital setting are protective against the development of opioid use disorder at later time points following hospital discharge [50]. Although both short- and long-term use of antibiotics drastically disrupts the gut microbiota, the microbial community may retain more beneficial or pathogenic bacteria [51,52,53]. The extent of antibiotic treatment in modifying gut bacteria is dependent on the class, pharmacokinetics, pharmacodynamics, and range of action, as well as their dosage, duration, and route of administration [53]. We found that our antibiotic treatment induced a significant depletion in Firmicutes and an increase in *Proteobacteria*, *Tenericutes*, and *Verrucomicrobia*. Proteobacteria are presumed to be an inflammatory microbe group and are elevated in diseased states [52,54]. Prior studies also observed shifts in *Firmicutes* and *Bacteroidetes*, the gut’s two dominant phyla, from antibiotic use [52,55].

The inclusion of both male and female animals in our study broadens the field of opioid research, as there is limited literature on sex differences in opioid abuse and even less when focused on gut microbial studies, which is concerning given the existing correlations between several addiction-related behaviors and the microbiome specific to sex [56]. We see that gut bacteria depletion via antibiotic treatment enhances fentanyl self-administration in males, but not females, at the lowest schedule of reinforcement (i.e., FR1), but there are no sex differences at higher-order schedules of reinforcement. Prior studies report lower drug use in females versus males [57], while others find greater self-administration and vulnerability to addiction in females [58,59,60,61]. The variation in these outcomes highlights the necessity for additional studies on gut microbiota, sex hormones, and drug-related behavior.

Our study is limited by the lack of assessment of antibiotic’s impact on fentanyl metabolism or overall locomotor activity, although previous work shows no significant influence of antibiotic treatment on cocaine or morphine metabolism in mice [24,28]. Further, non-reinforced responses (i.e., inactive nose poke hole) control for non-specific drug effects, including locomotor activity. Our limited number of collected fecal samples precluded an in-depth analysis of microbial interactions between fentanyl self-administration and antibiotic treatment, so future studies should evaluate these potential drug interactions. Furthermore, a causal relationship between gut microbiota and drug self-administration needs to be established. Future studies confirming gut bacterial changes with SCFA supplementation and/or evaluating microbial replacement via fecal microbiota transplantation may provide a direct connection between the gut microbiome and drug-related behavior.

Given the connections between gut microbiota and stress, mood, psychiatric disorders, and behavior, evaluating the role of gut microbiota in fentanyl use is a unique approach that could lead to new paths for the treatment of addiction. By identifying the gut-brain axis role in fentanyl use, our research has the potential to significantly progress our understanding of the mechanisms influencing the clinical use of opioids and addiction. One potential mechanism mediating the enhancement of fentanyl IVSA in males with depleted gut bacteria compared to control animals may be through neuroinflammation. Prior work showed that manipulation of the gut microbiota alters microglia morphology, a measure of neuroinflammation [27]. Neuroinflammation disrupts the function and projections of dopaminergic neurons within reward-related regions in the brain, leading to decreased mesolimbic dopaminergic activity and dysregulated reward [62]. Although our present study is observational, future mechanistic findings may uncover direct gut-to-brain pathways or intermediate modulators, such as neuroinflammation, microbial metabolites, gut peptides, and neurotransmitters.

## 4. Materials and Methods

Animals: Adult male and female Sprague Dawley rats (8 weeks of age) were obtained from Charles River (San Diego, CA, USA) and acclimated to our vivarium at least 7 days prior to experimentation. Animals were pair-housed in a humidity and temperature-controlled room on a 12 h light-dark cycle (lights on at 0700). Two separate experiments were run in this study, denoted as Antibiotic Treatment (Table 1 and Table 2) and Short-Chain Fatty Acid Supplementation (Table 3). 37 total animals were used in the antibiotic treatment experiment (20 males and 17 females) and 5 animals were excluded due to failure of catheter patency (Table 1). Fecal samples from a subset of the males were collected for analysis (Table 2). 49 total animals, all males, were used in the short-chain fatty acid supplementation experiment and 14 animals were excluded (8 due to failure of catheter patency and 6 outliers determined by box-and-whisker plots) (Table 3). All animal procedures were approved by the Institutional Animal Care and Use Committee (IACUC protocol number AUP-21-022) at the University of California Irvine and performed in accordance with the Association for Assessment and Accreditation of Laboratory Animal Care.

Antibiotic and Short-Chain Fatty Acid (SCFA) Treatment: Animals were randomly assigned to treatment groups using a random sequence generator. An antibiotic cocktail (2 g/L neomycin, 0.5 g/L bacitracin, 0.2 g/L vancomycin) was mixed in drinking water and provided ad libitum. SCFAs were mixed in drinking water in the following concentrations: 67.5 mM acetate, 40 mM butyrate, and 25.9 mM propionate (Sigma Aldrich, St. Louis, MO, USA). Water bottles were changed every 2 days and weighed daily to ensure the intake of antibiotics. The selected agents, doses, and treatment duration were adapted from Kiraly et al., 2016 [24]. Catheter Implantation and Drug Self-Administration: The methodology for catheterization surgery in preparation for fentanyl intravenous self-administration (IVSA) was described previously [33]. Rats were anesthetized with Equithesin (0.35 mL/100 g, intraperitoneal) and administered carprofen (4 mg/kg, subcutaneous) (Patterson Veterinary, Greeley, CO, USA) for post-operative analgesia. Animals were given at least 2 days to recover from surgery before fentanyl self-administration (see Ren and Lotfipour, 2022 for detailed methods) [33]. Briefly, fentanyl solutions were prepared using aqueous fentanyl citrate (Patterson Veterinary, Greeley, CO, USA) and sterile saline. All animals self-administered fentanyl at 1.25 μg/kg/infusion during daily 2 h sessions on an escalating schedule of reinforcement (i.e., 5 days at a fixed ratio (FR) 1, 2 days at FR2, 2 days at FR5, and 2 days at progressive ratio (PR)). This schedule of reinforcement has been previously established based on the time needed to acquire self-administration [63,64]. Animals self-administered fentanyl in individual chambers by poking their nose into a reinforced “nose poke” hole, which issued a cue light and delivery of one drug infusion with each response requirement met (e.g., one nose poke for one infusion at FR1, five nose pokes for one infusion at FR5, and a logarithmic increase within one session at PR). Nose pokes at a second non-reinforced hole resulted in no consequence, but the response was recorded to control for non-specific drug effects. Data were collected by a multichannel computer system (Med Associates, St. Albans, VT, USA). Catheter patency was tested after the final session on each schedule of reinforcement via i.v. administration of propofol (0.1 mL) (Zoetis, Parsipanny, NJ, USA). Data were discarded from 5 out of 37 animals not demonstrating rapid anesthesia.

Experimental Timeline (Antibiotic Treatment): Rats were treated with water or antibiotics via their home water bottles and remained on the same treatment for the duration of the experiment. One week after antibiotic treatment, rats underwent catheter implantation surgery in preparation for fentanyl IVSA and began self-administration after 2 days of recovery. One fecal sample was collected from each animal before surgery to confirm the knockdown of gut bacteria prior to IVSA, and again at the end of the experiment to ensure the maintenance of bacterial depletion (Figure 6).

Experimental Timeline (SCFA Supplementation): Animals were randomly assigned to the following experimental groups: water/water, water/SCFA, ABX/water, or ABX/SCFA. Rats were treated with water or antibiotics via their home water bottles for 3 days, and then the addition of either water or SCFA was added to the same bottle. Rats were maintained on the same treatment for the remainder of the experiment. One week after the start of antibiotic or water treatment, rats were implanted with catheters in preparation for fentanyl IVSA and began self-administration after 2 days of recovery (Figure 7).

16S Sequencing: Fecal samples from animals were collected in individual 1.7 mL Eppendorf tubes prefilled with DNA/RNA Shield (Zymo Research, Irvine, CA, USA) and stored at −80 °C until processing. Bacterial genomic DNA from all samples was isolated using the Zymobiomics DNA Mini Kit in a 96-well format (Zymo Research, Irvine, CA, USA). The genomic DNA was used to target the 16S rRNA gene. 16S rRNA amplicon PCR was performed targeting the V4–V5 region using the Earth Microbiome Project primers (515F (barcoded) and 926R) [65,66]. The library was sequenced at the University of California Irvine’s Genomics High Throughput Facility using Illumina MiSeq v3 (600 cycles) with a PE300 sequencing length. Sequencing resulted in 12M single-end reads (forward) passing filter of which 11% are PhiX with a >Q30 = 85%. The raw forward sequences were imported into QIIME2 (version 2020.8). After the initial sample quality check and trimming (DADA2 in QIIME2), there were 9.1 M single-end non-chimeric reads, which were used for further analysis. From the sequences, the first 5 bp were trimmed and truncated at 243 bp. The sequences were assigned a taxonomic classification using the August 2013 greengenes database (greengenes.secondgenome.com, accessed on 14 May 2021), trained with the full-length 16S gene region supplied by QIIME2.

Microbiome Analysis: Sequence data were exported from QIIME2 and integrated within R Studio. Within R Studio, we rarefied the feature table (organized by exact sequence variants) to the same sequencing depth (rarefaction depth = 7200, based on the lowest read distribution), plotted Shannon diversity index values, and created a distance matrix. Statistical comparison of the communities was performed using Tukey’s HSD, PERMANOVA, and multivariate ANOVA.

Statistical Analysis: Behavioral data were analyzed with JMP (SAS Institute, Cary, NC, USA). Each self-administration schedule of reinforcement (FR1, FR2, FR5, PR) was analyzed separately with multivariate ANOVA on sex, treatment, and response, with repeated measures on response (reinforced vs. non-reinforced). Any main effects were further analyzed using Bonferroni-corrected paired (response) or unpaired (drug) *t*-test post hoc comparisons.

## Figures and Tables

**Figure 1 ijms-24-00409-f001:**
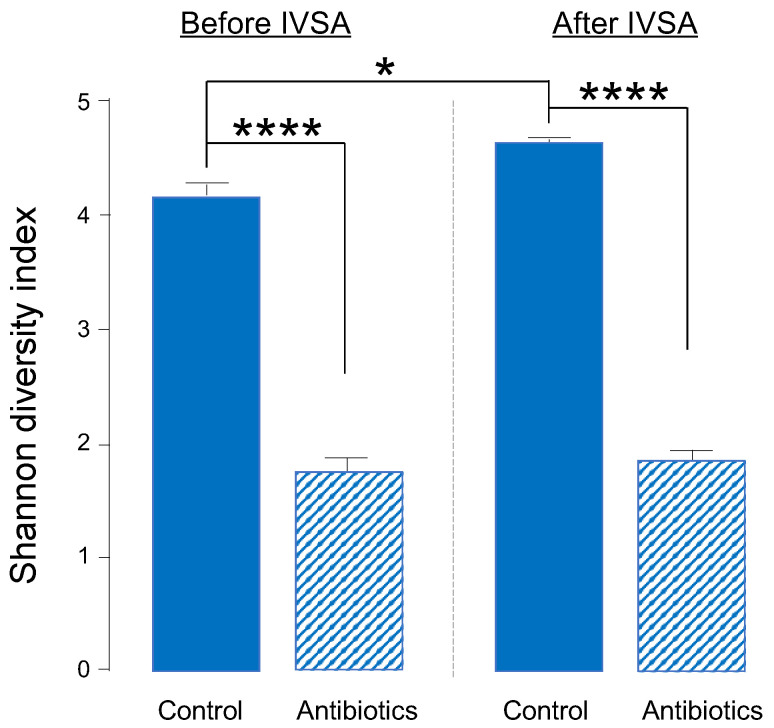
Shannon diversity index values from fecal samples before (left of dashed line) and after (right of dashed line) fentanyl intravenous self-administration (IVSA) in males treated with water only (solid bars) or water with antibiotics (dashed bars). Error bars represent S.E.M. **** *p* < 0.0001, * *p* < 0.05, *n* = 4–8/group.

**Figure 2 ijms-24-00409-f002:**
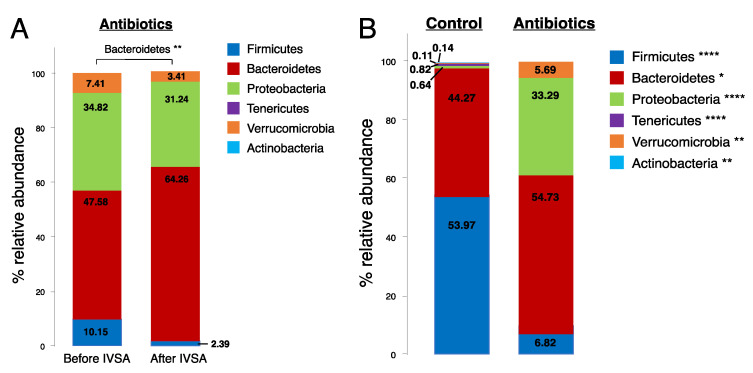
Alpha diversity at the phylum level in antibiotic-treated and control males. (**A**) Percent relative abundance of *Firmicutes* (dark blue), *Bacteroidetes* (red), *Proteobacteria* (green), *Tenericutes* (violet), *Verrucomicrobia* (orange), and *Actinobacteria* (light blue) in antibiotic-treated animals before and after fentanyl intravenous self-administration (IVSA). ** *p* < 0.01, *n* = 6–8/group. (**B**) Percent relative abundance of *Firmicutes*, *Bacteroidetes*, *Proteobacteria*, *Tenericutes*, *Verrucomicrobia*, and *Actinobacteria* in controls and antibiotic-treated animals. Data are collapsed by fecal collection timepoint. Asterisks (*) next to each phylum listed in the legend represent significant differences between controls and antibiotic treatment. **** *p* < 0.0001, ** *p* < 0.01, * *p* < 0.05, *n* = 9–14/group.

**Figure 3 ijms-24-00409-f003:**
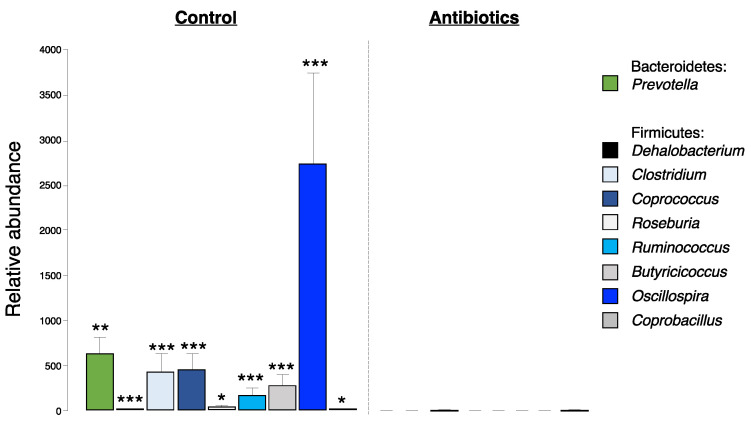
Alpha diversity at the genus level in control and antibiotic-treated males. Relative abundance of the genera *Prevotella*, *Dehalobacterium*, *Clostridium*, *Coprococcus*, *Roseburia*, *Ruminococcus*, *Butyricicoccus*, *Oscillospira*, and *Coprobaccilus* in controls and antibiotic-treated animals. Asterisks (*) above bars represent significant differences between controls and antibiotic treatment. Data are collapsed by fecal collection timepoint. Error bars represent S.E.M., *** *p* < 0.001, ** *p* < 0.01, * *p* < 0.05, *n* = 9–14/group.

**Figure 4 ijms-24-00409-f004:**
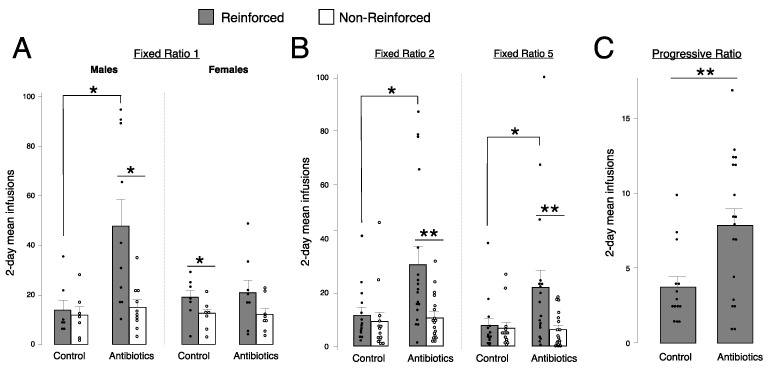
Fentanyl intravenous self-administration in controls and antibiotic-treated animals. All animals self-administered a fentanyl dose of 1.25 μg/kg/infusion. (**A**) Mean number of infusions across days 4–5 at fixed ratio (FR) 1 in males and females. Infusions were administered via nose pokes and are reported as reinforced (filled bars and circles) and non-reinforced (empty bars and circles) responses. Error bars represent S.E.M., * *p* < 0.05, *n* = 7–10/group. (**B**) Mean number of infusions across 2 days at FR2 and 2 days at FR5, collapsed by sex, for reinforced (filled bars and circles) and non-reinforced (empty bars and circles) nose poke responses. Error bars represent S.E.M., ** *p* < 0.01, * *p* < 0.05, *n* = 14–18/group. (**C**) Mean number of infusions across 2 days at progressive ratio, collapsed by sex. Error bars represent S.E.M., ** *p* < 0.01, *n* = 14–18/group.

**Figure 5 ijms-24-00409-f005:**
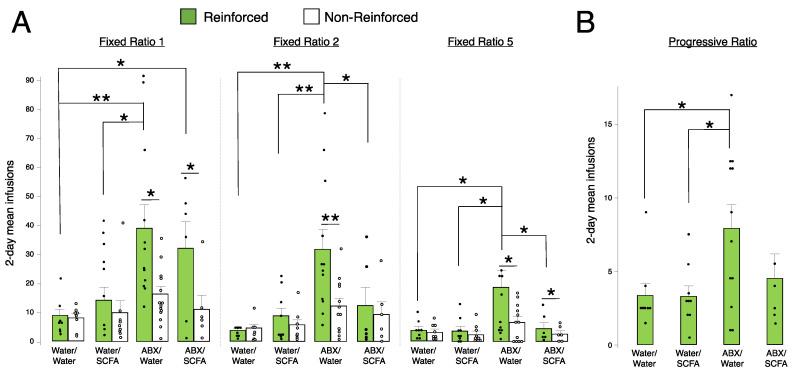
Fentanyl intravenous self-administration after short-chain fatty acid (SCFA) supplementation. All animals self-administered a fentanyl dose of 1.25 μg/kg/infusion and were treated with water, SCFA, antibiotics (ABX), or ABX plus SCFA. (**A**) Mean number of infusions across days 4–5 at fixed ratios 1, 2, and 5. Infusions were administered via nose pokes and are reported as reinforced (filled bars and circles) and non-reinforced (empty bars and circles) responses. Error bars represent S.E.M., ** *p* < 0.01, * *p* < 0.05, *n* = 6–12/group. (**B**) Mean number of infusions across 2 days at progressive ratio. Error bars represent S.E.M., * *p* < 0.05, *n* = 6–12/group.

**Figure 6 ijms-24-00409-f006:**
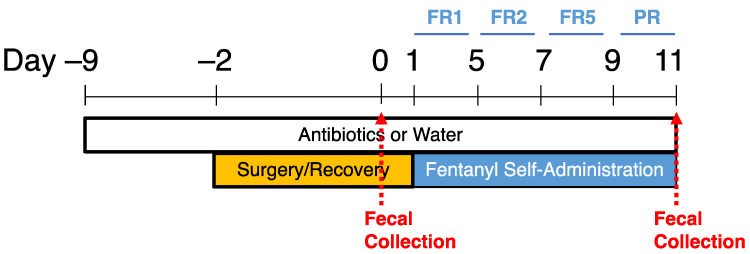
Experimental timeline for antibiotic treatment, fentanyl self-administration, and fecal sample collection. Animals were provided drinking water plus antibiotics or drinking water only (controls) and maintained their assigned treatment throughout the entire experiment. Following one week of antibiotic or water treatment, animals were implanted with intravenous catheters and given 2 full days to recover from surgery before starting self-administration. Animals self-administered fentanyl at fixed ratio (FR) 1 for 5 days, FR2 for 2 days, FR5 for 2 days, and progressive ratio (PR) for 2 days, for 11 days total. One fecal sample was collected from each animal the day before self-administration (day 0) and the last day of PR (day 11).

**Figure 7 ijms-24-00409-f007:**
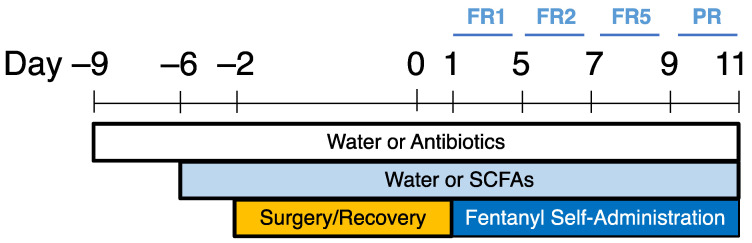
Experimental timeline for short-chain fatty acid (SCFA) supplementation and fentanyl self-administration. Animals were provided drinking water plus antibiotics or drinking water only for 3 days. SCFA or vehicle (water) was added to the treatment and rats were maintained on their assigned treatment throughout the entire experiment. Following one week of antibiotic treatment, animals were implanted with intravenous catheters and given 2 full days to recover from surgery before starting self-administration. Animals self-administered fentanyl at fixed ratio (FR) 1 for 5 days, FR2 for 2 days, FR5 for 2 days, and progressive ratio (PR) for 2 days, for 11 days total.

**Table 1 ijms-24-00409-t001:** Animal numbers per treatment group for fentanyl intravenous self-administration in antibiotic treatment.

	Control	Antibiotics
Males	7	10
Females	7	8

**Table 2 ijms-24-00409-t002:** Animal numbers per treatment group for fecal microbiome analysis before and after fentanyl intravenous self-administration (IVSA). Only males were used for microbiome analysis due to limited samples.

	Control	Antibiotics
Before IVSA	5	8
After IVSA	4	6

**Table 3 ijms-24-00409-t003:** Animal numbers per treatment group for fentanyl intravenous self-administration in short-chain fatty acid (SCFA) supplementation with or without antibiotics (ABX).

	Water/Water	Water/SCFA	ABX/Water	ABX/SCFA
Males	8	9	12	6

## Data Availability

The data presented in this study are available on request from the corresponding author. The data are not publicly available due to the privacy of funding sources and ongoing related research.

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
