# Peer review of "Antibiotic Knockdown of Gut Bacteria Sex-Dependently Enhances Intravenous Fentanyl Self-Administration in Adult Sprague Dawley Rats"

_ijms, 2022, doi:10.3390/ijms24010409_

Round 1

Reviewer 1 Report (New Reviewer)

In the submitted manuscript, Ren and Lotfipour show data contributing to the hypothesis of interaction between gut microbiota and the brain as a model of opioid drug abuse. Although the manuscript does not uncover a new mechanism of action, it serves to strengthen the emerging associations. I consider the manuscript to meet the criteria of the journal and after an extensive revision could be accepted for publication. Below I list some major and minor points I would suggest the authors to improve upon.

Major points:

1. At the end of introduction, the aims of the study were not clearly specified. For instance, there was no mention of determining the abundance and diversity of gut microbiota. Although it may not have been a primary outcome, it could have been mentioned.

2. Chapter 2.1 of the results is presented in the text in such a way which may be confusing. For example, I would suggest the authors to refer to Figure 1A after the sentence “In addition to a main effect of treatment…” as it is with this sentence where the description of Figure 1A ends. From the next sentence (starting with “upon a deeper analysis”), it is not clear which groups are being compared, as Figure 1B shows 4 groups (divided based on the administration of antibiotics and based on the time of IVSA). Additionally, while describing the results of Figure 1B and 1C in the text, the authors use the term “significant differences” without mentioning the actual difference (if it was lower or higher). There is a line denoting a 2-star significance of Bacteroidetes but on the right of Figure 1B, there is a legend showing various other significances. Neither from the text nor from the graph it is clear between which groups the significances are (a simple line connecting the groups would be enough). It seems to me that the authors tried to put as much data into one graph as possible, however, unless necessary, I would advise to separate Figure 1B into at least two figures, one showing the 2-star difference in Bacteroidetes between certain groups and the other showing all the other significances found between other two groups. If possible, the number values of the percentages of relative abundance of each of the six bacterial phyla shown in Figure 1B could be added and possibly mentioned in the results section as well (the authors show the graph and degrees of significance without the actual numbers). In addition, the graph type used in Figure 1B makes the differences among the various bacterial phyla less prominent.

3. Figure 1C looks distorted so that the individual names of the taxa cannot be read clearly. Apart from that, I would suggest the authors to create a bar graph in order to visualise the differences (instead of just depicting it with an arrow). As the abbreviation “IVSA” is explained in the legend of Figure 1, it makes sense also to explain “ABX” (which I presume should mean “antibiotics”).

4. Figure 2A is not very clear. The y axis states “2-day mean infusions”, so an average number of fentanyl administrations but the figure legend states “nose poke responses are reported as reinforced and non-reinforced”. From the overall description, it appears as if the graph depicted one set of data while the legend of the figure described another one (e.g. the number of infusions and the number of nose poke responses) which may be rather confusing to the common reader. I believe it would be beneficial if authors could explain in the results chapter the relationship between the number of infusions and nose poke responses.

5. In chapter 2.2, the authors wrote that they analysed a 2-day average of days 4 and 5 because of no difference at FR1. It was not stated in the manuscript whether all of the fixed ratios were being compared or only some of them and why. In addition, the chapter lacked coherence, as the next sentence claimed that the last two days were selected which should be the days 9-11 belonging to the progressive ratio (according to Figure 3). After that, the focus returned to FR1, writing about sex differences. I would suggest the authors to order the description of results so that it follows some logical order, e.g. fixed ratios starting at FR1 and going towards progressive ratio. In Figure 2C, it may be useful to point out to the reader that the y axis of progressive ratio is far lower compared to FR1, FR2 or FR5.

Minor points:

1. As a person unfamiliar with fixed ratios in this kind of study, I would appreciate an explanation in the discussion regarding why FR1 lasted for 4 days (day 1 – 5) while the other two only for 2 days (including the progressive ratio). (If this time schedule was developed by some other research group in the past as a suitable experimental design for this kind of study, one sentence with a citation is enough.) Likewise, I would also appreciate a definition of the term “reinforced” and “non-reinforced” to understand the difference between a reinforced and non-reinforced response in more detail.

2. In all those figures, tables and legends to figures and tables where applicable, I would advise to change “water” denoting the control group and use “control” instead.

3. I noticed different amounts of animals stated in different figures. Does it mean that different animals were used for the experiments (fentanyl administration and gut microbiota analysis, for example)? Apart from that, Table 2 shows the amount of stool samples from males used for gut microbiota analysis. The total of used males was 20 but in the table, the sum of all numbers per group is 23. Is it possible that those 3 males were excluded due to the failure of catheter patency as stated in the text above?

4. In the discussion, I would suggest adding at least two citations in the sentence starting with “While prior groups have reported…” (line 136) so that it is clear which groups and which publications the authors mean. Alternatively, if those citations which are at the end of the previous sentence apply, they could be moved to the end of the second sentence.

5. Line 169: “Proteobacteria is presumed to be an inflammatory microbe…” here, plural would be in order or, it could be spoken about Proteobacteria in singular as a taxon or phylum.

6. Several references lack consistency: in some references, the name of the journal is listed as an abbreviation while in others the full name is written (Scientific Reports, for example). References number 6, 36 and 51 have listed “null” instead of journal name. Please check the citation manager if the names of the journals are not missing in those references. Additionally, reference number 64 lacks publication year and publication years of references 8 and 57 are not written in bold (as opposed to all the others).

Author Response

Point 1: At the end of introduction, the aims of the study were not clearly specified. For instance, there was no mention of determining the abundance and diversity of gut microbiota. Although it may not have been a primary outcome, it could have been mentioned.

Response 1: We have now added the objective to measure alpha diversity.

Point 2: Chapter 2.1 of the results is presented in the text in such a way which may be confusing. For example, I would suggest the authors to refer to Figure 1A after the sentence “In addition to a main effect of treatment…” as it is with this sentence where the description of Figure 1A ends. From the next sentence (starting with “upon a deeper analysis”), it is not clear which groups are being compared, as Figure 1B shows 4 groups (divided based on the administration of antibiotics and based on the time of IVSA). Additionally, while describing the results of Figure 1B and 1C in the text, the authors use the term “significant differences” without mentioning the actual difference (if it was lower or higher). There is a line denoting a 2-star significance of Bacteroidetes but on the right of Figure 1B, there is a legend showing various other significances. Neither from the text nor from the graph it is clear between which groups the significances are (a simple line connecting the groups would be enough). It seems to me that the authors tried to put as much data into one graph as possible, however, unless necessary, I would advise to separate Figure 1B into at least two figures, one showing the 2-star difference in Bacteroidetes between certain groups and the other showing all the other significances found between other two groups. If possible, the number values of the percentages of relative abundance of each of the six bacterial phyla shown in Figure 1B could be added and possibly mentioned in the results section as well (the authors show the graph and degrees of significance without the actual numbers). In addition, the graph type used in Figure 1B makes the differences among the various bacterial phyla less prominent.

Response 2: We have incorporated these suggestions. Namely, we have rewritten some points for clarity, split Figure 1 into 3 figures, and added number values to the stacked bar graph.

Point 3: Figure 1C looks distorted so that the individual names of the taxa cannot be read clearly. Apart from that, I would suggest the authors to create a bar graph in order to visualise the differences (instead of just depicting it with an arrow). As the abbreviation “IVSA” is explained in the legend of Figure 1, it makes sense also to explain “ABX” (which I presume should mean “antibiotics”).

Response 3: The previous image (arrows to indicate decrease or increase) has now been replaced with a bar graph to better represent the taxa differences.

Point 4: Figure 2A is not very clear. The y axis states “2-day mean infusions”, so an average number of fentanyl administrations but the figure legend states “nose poke responses are reported as reinforced and non-reinforced”. From the overall description, it appears as if the graph depicted one set of data while the legend of the figure described another one (e.g. the number of infusions and the number of nose poke responses) which may be rather confusing to the common reader. I believe it would be beneficial if authors could explain in the results chapter the relationship between the number of infusions and nose poke responses. In chapter 2.2, the authors wrote that they analysed a 2-day average of days 4 and 5 because of no difference at FR1. It was not stated in the manuscript whether all of the fixed ratios were being compared or only some of them and why. In addition, the chapter lacked coherence, as the next sentence claimed that the last two days were selected which should be the days 9-11 belonging to the progressive ratio (according to Figure 3). After that, the focus returned to FR1, writing about sex differences. I would suggest the authors to order the description of results so that it follows some logical order, e.g. fixed ratios starting at FR1 and going towards progressive ratio. In Figure 2C, it may be useful to point out to the reader that the y axis of progressive ratio is far lower compared to FR1, FR2 or FR5.

Response 4: We have added the relationship between infusions and nose poke responses (in Methods/Materials), edited for clarification about the last two days on FR1 and not the entire IVSA timeline (Results), and made a note about the y-axis being lower on progressive ratio vs. FR1, FR2, and FR5 (Results).

Point 5: As a person unfamiliar with fixed ratios in this kind of study, I would appreciate an explanation in the discussion regarding why FR1 lasted for 4 days (day 1 – 5) while the other two only for 2 days (including the progressive ratio). (If this time schedule was developed by some other research group in the past as a suitable experimental design for this kind of study, one sentence with a citation is enough.) Likewise, I would also appreciate a definition of the term “reinforced” and “non-reinforced” to understand the difference between a reinforced and non-reinforced response in more detail.

Response 5: We have now added 2 references in the methods that have used fixed and progressive ratio schedules of reinforcement, both of which determine the length of time necessary to acquire drug self-administration.

Point 6: In all those figures, tables and legends to figures and tables where applicable, I would advise to change “water” denoting the control group and use “control” instead.

Response 6: This suggestion has been updated.

Point 7: I noticed different amounts of animals stated in different figures. Does it mean that different animals were used for the experiments (fentanyl administration and gut microbiota analysis, for example)? Apart from that, Table 2 shows the amount of stool samples from males used for gut microbiota analysis. The total of used males was 20 but in the table, the sum of all numbers per group is 23. Is it possible that those 3 males were excluded due to the failure of catheter patency as stated in the text above?

Response 7: The animal numbers are different due to stool samples analyzed from a smaller subset of the animals used in the self-administration experiments. We have now added this point in the methods and results. In addition, the number of stool samples shown in Table 2 is divided by treatment group and collection timepoint. We aimed to collect 2 samples per animal (one before and one after fentanyl self-administration), but only males demonstrating catheter patency had samples collected after self-administration, which is why the stool numbers are different for before vs. after self-administration.

Point 8: In the discussion, I would suggest adding at least two citations in the sentence starting with “While prior groups have reported…” (line 136) so that it is clear which groups and which publications the authors mean. Alternatively, if those citations which are at the end of the previous sentence apply, they could be moved to the end of the second sentence.

Response 8: We have now moved the references.

Point 9: Line 169: “Proteobacteria is presumed to be an inflammatory microbe…” here, plural would be in order or, it could be spoken about Proteobacteria in singular as a taxon or phylum.

Response 9: This has now been corrected.

Point 10: Several references lack consistency: in some references, the name of the journal is listed as an abbreviation while in others the full name is written (Scientific Reports, for example). References number 6, 36 and 51 have listed “null” instead of journal name. Please check the citation manager if the names of the journals are not missing in those references. Additionally, reference number 64 lacks publication year and publication years of references 8 and 57 are not written in bold (as opposed to all the others).

Response 10: The references have now been reviewed to match formatting. All references are also updated using journal abbreviations.

Reviewer 2 Report (New Reviewer)

It was interesting to read the research article by Ren and Lotifipour. It emphasized the significant connection between the gut microbiome and opioid usage. The article is orderly and well-written. However, in Figure 1C, under phylum the words are illegible. Could you please change it to more readable fonts?

Author Response

Point 1: In Figure 1C, under phylum the words are illegible. Could you please change it to more readable fonts?

Response 1: We have now changed this figure for better readability.

Round 2

Reviewer 1 Report (New Reviewer)

All my previous comments were sufficiently addressed. I have no additional points to list.

This manuscript is a resubmission of an earlier submission. The following is a list of the peer review reports and author responses from that submission.

Round 1

Reviewer 1 Report

1. There are still some spelling and grammatical problems in this paper, please re-check it.

2. As the authors mentioned in Results that a significant difference in Shannon diversity before and after fentanyl self-administration in antibiotic-treated animals was driven by Bacteroidetes, Bacteroidetes may play an important role in response (Shannon diversity index and even infusions in the schedules of reinforcement) to fentanyl self-administration upon antibiotic treatment. However, there is little knowledge about it in the Discussion. Please give more details about it.

3. In the end of the first paragraph, in line 30-31, the authors said “In this study, we show gut bacteria as a potential mechanism underlying fentanyl intravenous self-administration (IVSA) in adult Sprague Dawley rats.” It is inappropriate because of two points. Firstly, according to the following descriptions, the “potential mechanism” should be the bidirectional communication between the gut bacteria and host. The description of this sentence should be improved. Secondly, this sentence pointed out the object - gut bacteria, while there was no any information before to draw out this object. Please do some modifications on it.

4. In the present study, the authors highlighted the bidirectional communication between the brain and gut microbiota, however, no direct links between the gut microbe and brain were exhibited. Furthermore, more evidences about “bidirectional communication” should be provided. More data or literature search are needed.

5. In the first paragraph of the Discussion section, in line 119-120, the authors proposed their prospects, but it is more suitable to give prospects based on results of the research in the end of the paper. 

Author Response

Point 1: There are still some spelling and grammatical problems in this paper, please re-check it.
Response 1: We revised the previous draft and made edits to spelling, punctuation, and verb tenses.

Point 2: As the authors mentioned in Results that a significant difference in Shannon diversity before and after fentanyl self-administration in antibiotic-treated animals was driven by Bacteroidetes, Bacteroidetes may play an important role in response (Shannon diversity index and even infusions in the schedules of reinforcement) to fentanyl self-administration upon antibiotic treatment. However, there is little knowledge about it in the Discussion. Please give more details about it.
Response 2: We expanded the Discussion to add more information about Bacteroidetes.

Point 3: In the end of the first paragraph, in line 30-31, the authors said “In this study, we show gut bacteria as a potential mechanism underlying fentanyl intravenous self-administration (IVSA) in adult Sprague Dawley rats.” It is inappropriate because of two points. Firstly, according to the following descriptions, the “potential mechanism” should be the bidirectional communication between the gut bacteria and host. The description of this sentence should be improved. Secondly, this sentence pointed out the object - gut bacteria, while there was no any information before to draw out this object. Please do some modifications on it.
Response 3: We have now rewritten the end of this paragraph to improve clarification and transition.

Point 4: In the present study, the authors highlighted the bidirectional communication between the brain and gut microbiota, however, no direct links between the gut microbe and brain were exhibited. Furthermore, more evidences about “bidirectional communication” should be provided. More data or literature search are needed.
Response 4: We added references and elaborated on data from previously cited studies to provide more information on the direct and bidirectional communication between the brain and gut bacteria.

Point 5: In the first paragraph of the Discussion section, in line 119-120, the authors proposed their prospects, but it is more suitable to give prospects based on results of the research in the end of the paper. 
Response 5: Thank you for this suggestion. We moved these mechanistic hypotheses to the end of the Discussion.

Reviewer 2 Report

The authors in this manuscript described the effect of antibiotic knockdown of gut bacteria which enhances intravenous fentanyl self-administration in Dawley Rats. Study has many contradictions to previously shown reports as mentioned by the authors. The results presented are not highly convincing to support the current data. The following are the major concerns:

1.       As described by authors, knockdown of gut bacteria influences enhancement of self-administration in males, but the report fails to explain how does that happen? The authors should atleast propose any mechanism of this finding.

2.       There are a lot of discrepancies shown in contradiction to old studies.

3.       Title should be modified to ‘Adult Sprague Dawley Male Rats’ since the effect is observed only in Males and not female Rats.

4.       Substantial improvement is grammar is required.

5.       Data shown is not enough. It simply demonstrates an observational study.

Author Response

Point 1: As described by authors, knockdown of gut bacteria influences enhancement of self-administration in males, but the report fails to explain how does that happen? The authors should at least propose any mechanism of this finding.
Response 1: We added our proposed mechanism at the end of the Discussion.

Point 2: There are a lot of discrepancies shown in contradiction to old studies.
Response 2: We included a statement in our discussion to highlight the methodological differences in our study compared to old studies.

Point 3: Title should be modified to ‘Adult Sprague Dawley Male Rats’ since the effect is observed only in Males and not female Rats.
Response 3: We have updated the title to reflect the sex differences in our study. The treatment effect is only in male rats at fixed ratio 1, but we see an enhancement in self-administration in both males and females at higher schedules of reinforcement.

Point 4: Substantial improvement is grammar is required.
Response 4: This updated draft includes edited punctuation and verb tenses.

Point 5: Data shown is not enough. It simply demonstrates an observational study.
Response 5: We acknowledged this point at the end of the manuscript and aim for this to move into a mechanistic study.

Round 2

Reviewer 1 Report

No

Reviewer 2 Report

Figures are too congested and hard to interpret to the readers.